# A Multicriteria Model for Estimating *Coffea arabica* L. Productive Potential Based on the Observation of Landscape Elements



**Jorge Eduardo F. Cunha** [1], **George Deroco Martins** [2], **Eusímio Felisbino Fraga Júnior** [2], **Silvana P. Camboim** [3] and **João Vitor M. Bravo** [4,*]

1   Insitute of Agricultural Sciences, Campus Monte Carmelo, Federal University of Uberlandia, Monte Carmelo 38500-000, Minas Gerais, Brazil; jorge.cunha@ufu.br
2   Insitute of Geography, Campus Monte Carmelo, Federal University of Uberlandia, Uberlandia 38500-000, Minas Gerais, Brazil; deroco@ufu.br (G.D.M.); eusimiofraga@ufu.br (E.F.F.J.)
3   Departament of Geomatics, Campus Polytechnic Center, Federal University of Parana, Curitiba 81530-900, Parana, Brazil; silvanacamboim@gmail.com
4   Insitute of Geography, Campus Santa Monica, Federal University of Uberlandia, Uberlandia 38408-100, Minas Gerais, Brazil
*   Correspondence: jvmbravo@ufu.br

**Abstract:** Understanding a crop's productive potential is crucial for optimizing resource use in agriculture, encouraging sustainable practices, and effectively planning planting and preservation efforts. Achieving precise and tailored management strategies is equally important. However, this task is particularly challenging in coffee cultivation due to the absence of accurate productivity maps for this crop. In this article, we created a multicriteria model to estimate the productive potential of coffee trees based on the observation of landscape elements that determine environmental fragility (EF). The model input parameters were slope and terrain shape data, slope flow power, and orbital image data (Landsat 8), allowing us to calculate the NDVI vegetation index. We applied the model developed to coffee trees planted in Bambuí, Minas Gerais, Brazil. We used seven plots to which we had access to yield data in a recent historical series. We compared the productivity levels predicted by the EF model and the historical productivity data of the coffee areas for the years 2016, 2018, and 2020. The model showed a high correlation between the calculated potential and the annual productivity. We noticed a strong correlation ($R^2$) in the regression analyses conducted between the predicted productive potential and the actual productivity in 2018 and 2020 (0.91 and 0.93, respectively), although the correlation was somewhat weaker in 2016 (0.85). We conclude that our model could satisfactorily estimate the yearly production potential under a zero-harvest system in the study area.

**Keywords:** coffee; crop estimation; productive potential; environmental fragility; sustainability



## 1. Introduction

Modern society desires sustainable agriculture that combines environmental responsibility with productivity, achieved through minimally invasive practices [1,2]. At the same time, this desire for evolution has encountered barriers in the accelerated population growth and the chaotic occupation of geographic space, putting the preservation of the Earth's natural resources at risk [3]. Indeed, it is necessary to develop strategies that enable sustainable production growth through knowledge and management of available resources, especially in areas where agriculture is practiced [4].

Precision agriculture (PA) has stood out as an essential ally in developing intelligent natural resource management strategies. In this modality, crops are managed according to the spatial and temporal variability of environmental factors [5]. The use of PA in

agricultural practices leads to the rational use of inputs, directly promoting sustainability [6]. Also, indirectly, PA provides a lower entry of chemical inputs into the crops, lowers machinery traffic, avoids soil compaction, and reduces the risks of water contamination and damage to the human health of workers involved with the application of chemical inputs [7].

Despite the technological advances in management practices, genetic improvement, use of biological inputs, which are productive pillars of agricultural systems, and the support of PA in increasing productivity, environmental variables such as climate, soils, and availability of water, are still dominant factors in productivity [8]. For example, regarding coffee cultivation, it is known that topography and soil type influence the water-holding capacity and the maintenance and availability of macro- and micronutrients [9–11]. These capacities are closely related to the productivity and quality of coffee trees [12–14]. Hence, it might be surmised that it would suffice to cartograph soil types in conjunction with localized meteorological data to ascertain—satisfactorily—the productive potential of coffee plantations [7]. However, soil mapping is an expensive task in its traditional format, making it impossible to cover large areas with maps produced with detailed scales [14].

It is important to stress that, in this context, we define productive potential as the inherent production capacity of a specific crop, considering its genetics, climate, soil quality, and management practices, without the influence of biotic and abiotic stresses [15,16]. Complementarily, productivity is the measure that relates to the quantity of a product produced in a given space [17]. Here, we adopted the measurement in sacks of processed coffee harvested per hectare for both concepts. When investigating the productive potential of a region or a crop, the professionals involved with this task will naturally adopt sustainable agriculture practices since they will plan management according to local capacities and conditions.

An alternative for mapping the productive potential of crops is based on the observation of covariates that allow for the prediction or measurement of parameters. These parameters could be registered by remote-sensing techniques [12,15]. In this case, it is necessary to observe variables related to soils, vegetation, relief, and landscape, as there is a close relationship between these factors and productivity. Here, we understand that it would be enough to interpret the connections between these landscape elements to determine a crop or region's productive potential with a certain degree of accuracy.

We emphasize that measuring the productive potential of agricultural areas allows for rational planning using inputs [3,18], mainly fertilizers since the fertilization of the coffee crop is commonly carried out based on the extraction of nutrients expected for each plot [19,20]. Such fertilizers, in excess, are harmful to the environment as they can pollute the water. In addition, the prior mapping of the productive potential of coffee trees allows for the conscious occupation of areas and the use of water, maximizing the use of natural resources. It would benefit environmental conservation, facilitating the maintenance and restoration preservation of the regional fauna and flora or even intercropping in places with less productive potential [5,21,22]. Specifically, multispectral images, digital elevation models, and enhancement techniques are essential for landscape reading and interpretation [13,17]. Therefore, researchers have the tools required to develop strategies that sustainably manage natural resources allied to increased crop productivity [7,18,23]. The objective of this study was to identify which variables allow us to know the productive potential of coffee trees and if it is possible to link the prediction of the productive potential of crops to the reading of the features of the natural landscape so that the conservation of natural resources is instigated.

Here, we created a model that allows for the estimation of the productive potential of coffee areas and the estimation of its annual dynamics associated with the vegetative state of the plant for coffee areas in a zero-harvest system using free spatial and multimodal data. We understand that it is possible to associate the productive potential of the coffee tree with variables related to topography (slope shapes, water flow power, and slope) and the plant (age and vigor) to create a prediction model for crop yields [13,14]. Such a modeling

process enables the rational planning of using inputs and natural resources and choosing areas with greater potential for exploration without risking conservation. Therefore, the model we created proved satisfactory in measuring the productive potential of the coffee plants we studied.

## 2. Materials and Methods

### 2.1. Study Area

The study area comprises seven plots, defined by the producer during planting and followed by the divisions in our work. The plots in the area are identified by the year of planting as follows: coffee 1999, with 30 hectares; 2000, with 30 hectares; 2002, with 32 hectares; 2003, with 20 hectares; 2004, with 40 hectares; 2005, with 49 hectares; and 2007, with 46 hectares, totaling 247 hectares of functional coffee area, located on the São Francisco farm, municipality of Bambuí-MG. To distinguish the stands, they were renamed here: A, B, C, D, E, F, and G, respectively, according to the chronological order of planting. Figure 1 shows the geographic region in which the rural property is located with the plots of the case study. The coffee areas highlighted as the study area use the rainfed planting system, with a spacing of 3.6 m between rows and 0.5 m between plants. Planting is preferably carried out in an east–west direction with a slight inclination to the north, except for coffee 2003 (D), where the planting took place in a southeast–northwest direction due to the slope inclination. The cultivars in the areas are Catucaí Amarelo 20/15 cv 479 for areas E and F, Tupi RN IAC 1669-13 for area C, and Catuaí Amarelo IAC cv. 62 for the other areas (A, B and D).

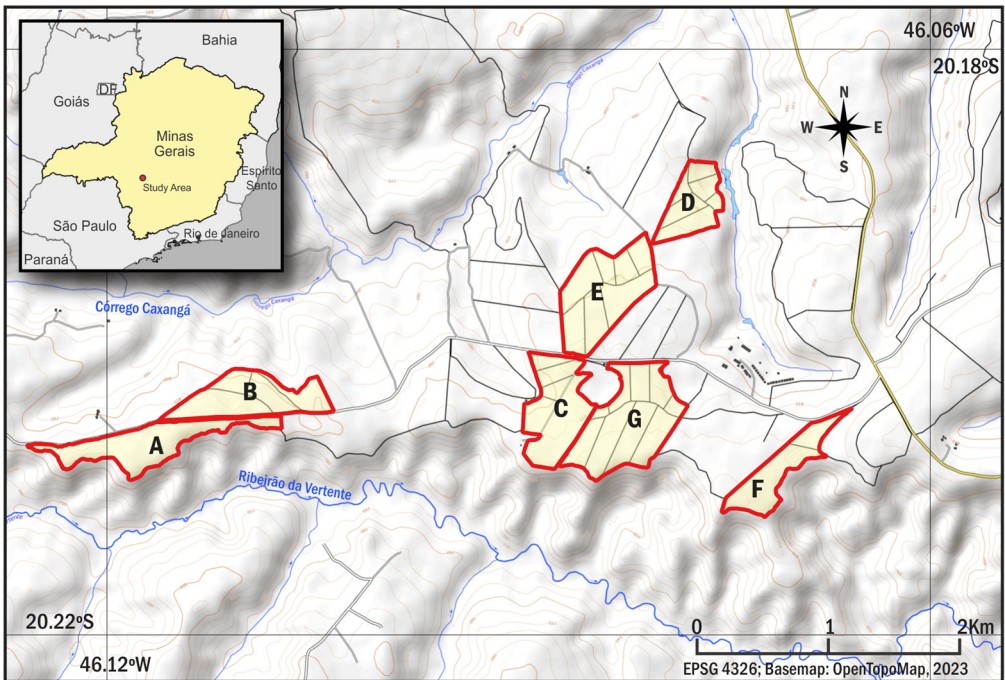

**Figure 1.** Location of the state of Minas Gerais, and in detail the study area and its respective plots: A, B, C, D, E, F and G.

The zero-harvest scheme involves pruning plagiotropic branches after each harvest and stripping orthotropic branches, resulting in no harvests in the following year. The second year sees a significant increase in crop production, effectively condensing two years of harvest into one. The proposed model in this study predicts crop years within this system, disregarding spectral behavior variations due to the biennial cycle and limiting its application to areas where zero-harvest crop management is practiced.

Regarding the *Urochloa ruziziensis* planting scheme, the entire system is composed of planting between the lines with the forage species, a grass with high mass production in the

shoot, and a robust root system. Brachiaria is planted in the harvest year, after pruning, at the beginning of the rains, and desiccation is carried out only after approximately 20 months of planting to clean the street for harvest, which provides great vegetation and seeding of the species in the area. It is important to emphasize this way of planting in the area because of the changes that must occur in the spectral response during remote sensing due to the system, compared to the response occurring in bush management and clean street management systems (Figure 2) [24,25].

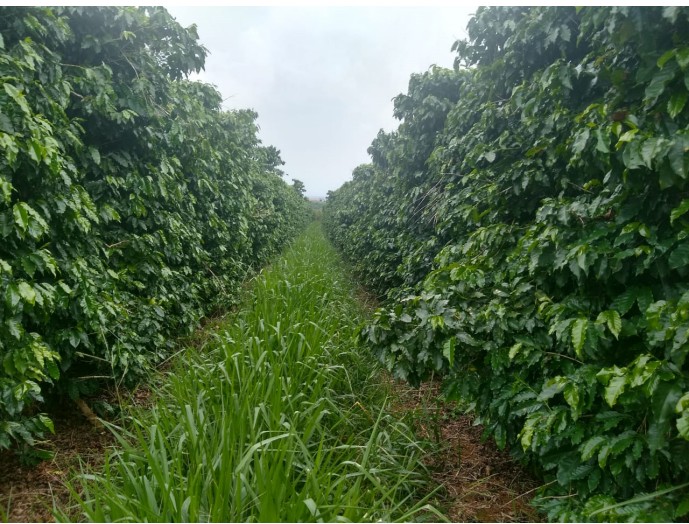

**Figure 2.** Coffee plantation intercropped with Urochloa ruziziensis. Postpruning year (crop 0) in the C area.

Soil correction is executed annually using dolomitic C limestone from the base saturation method on the cation exchange capacity (CEC) of the soil for liming and gypsum. Potassium correction (potash) and phosphorus (phosphate) in preplanting are also used. The entire correction procedure is carried out from the definition of management zones in precision agriculture, and fertilization is carried out based on the estimate of future harvest and crop extraction.

The study area is located in the São Francisco sedimentary basin region, within the morphological unit called Dissected Plateau of Bambuí, with altitudes between 650 and 850 m above sea level (average of 750 m). The region's relief is mostly smooth wavy to wavy, composed mainly of Cambisols. There is the occurrence of flat tops, where Latosols predominate and coffee cultivation occurs. The local climate, according to the Koppen classification, is Cwa, with medium temperatures, hot (>22 °C) and rainy summers, and medium temperatures (<18 °C) and dry winters. The average annual rainfall is approximately 1700 mm (Martins, 2013; Schaefer, 2013). Typical years were observed during the study periods without sudden variations in climatic variables.

### 2.2. Method

Here we follow the Ross (1994) proposal, with adaptations, to create a model for predicting the productive potential of coffee plantations. According to empirical observations, Ross (1994) assigned weights to different natural parameters, such as slope, type of vegetation, occupation of the site, and soil orders, and used simple algebra to generate a value that functions as an environmental fragility score. According to each case, other authors have widely reused this method with different parameters and equations [26–31]. Here, the model proposal was adapted to create productive potential scores. Each class value for the parameters analyzed and the values resulting from their combinations are dimensionless and assigned based on empirical knowledge of the areas studied. In this way, this model can only be built in other situations with previous data and specific site knowledge to create the class values.

In this way, we adapted Ross's proposal (1994) and applied the solution we created in a region with monitored coffee trees, which served as a case study to test the model. Consequently, when trying the model, we sought to predict the productive potential of coffee stands planted in Bambuí, MG, observing whether the adapted covariates would correlate with the productivity measured in 2016, 2018, and 2020.

While developing our model for estimating production potential, the input parameters were adapted according to the empirical observation of the coffee-plant response in the field and the available literature (Table 1). The flowchart in Figure 3 illustrates the process of obtaining the proposed prediction model, which may be replicated in other geographic contexts. We emphasize that all procedures were performed with QGIS software, and the remote sensing products were obtained from free platforms. Our proposal is a low-cost solution that can be replicated without significant investments.

**Table 1.** Data used in the study, types, sources, and references.

| Data | Type | Source | References |
|---|---|---|---|
| Digital Elevation Model | Raster Image | Topodata–Inpe | Valeriano [32] |
| Multispectral Image (NDVI) | Raster Image | Copernicus–ESA | |
| Spectral Image (Thermal) | Raster Image | USDA–NASA | |
| Slope Shape | Raster Image | Topodata–Inpe | Valeriano [32] |

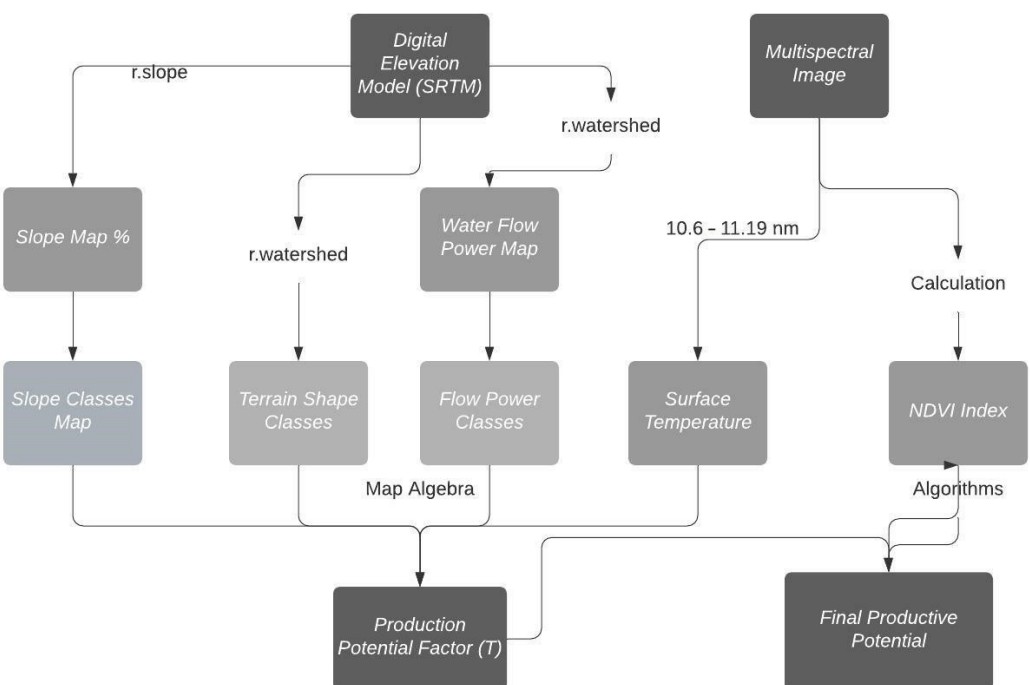

**Figure 3.** Flowchart of methodological processes of the proposed model.

Regarding the productive potential mapping process, we used the Digital Elevation Model (DEM) from Topodata [32] to obtain maps of the parameters: slope, water flow power, and slope shape. The QGIS tools r.slope and r.watershed were used to process the DEM for slope and water flow power. For slope shape, the product is directly available at Topodata; we also used the thermal band of the Thermal Infrared Sensor (TIRS) from Landsat 8. The data source and its types can be seen in Table 1. From these products, we generated the map of the potential factor of coffee production proposed in our work and named factor T (Equation (1)), representing the productive potential of the stands in the study area. To generate the map, we used the value of the averages of the classes of each parameter for each field based on descriptive zonal statistics. The analysis was carried out

by stand due to the unavailability of a reliable map of the spatial variability of productivity, a common problem of the coffee culture, mainly due to the difference in density between different stages of fruit maturation. Thus, the analysis was performed by plot, based on the actual production history obtained from drying and processing, carried out individually by plot within the farm itself. In any case, our model can help in the absence of the productivity map in the broad spectrum of its uses, such as directed replacement of nutrients, fertilization planning, investigation of spatial/temporal variability, and the formation of management zones. Again, the absence of a reliable map of yield variation is a common situation in coffee farming.

The classes defined for slope and temperature were numbered in ascending order (1—very bad, 2—poor, 3—average, 4—good, and 5—very good). In contrast, the classes for flow power and terrain shape were numbered in a decreasing way (1—very good, 2—good, 3—average, 4—bad, and 5—very bad) to generate two classes directly and two indirectly proportional, ending in Equation (1), and the numbers used are the medium of classes for each plot. The T value is the result of combining the class values in Table 1, which are in brackets after each class interval. These values have been allocated in Equation (1), resulting in the T class values.

$$T = (Declivity + Temperature)/(Flow + Shape) \tag{1}$$

Specifically, we created the hierarchical categories of Table 1 based on the knowledge that sites previously known for their history of high productivity in the study area were associated with specific classes of the observed variables and sites of low productivity. In addition to the observations, we used the literature on the relationship of these variables with coffee yield [13,14,33–35].

Thus, in assigning weights, we understand that slope is a variable with an inversely proportional relationship to coffee productivity, as reported in the literature [25]. However, we also realize that plane terrains generally have lower productivity when compared to slightly sloping reliefs. In addition to this, it is essential to point out that there may be specific occasions where this relationship may change according to the weathering process and the management of the production system, making it necessary to adjust the model to the geographic region where the method we have developed will be applied.

Regarding the land shape, there are studies relating the slopes' forms on which coffee trees are planted with productivity parameters and product quality [13,14]. Based on this finding, we mapped and analyzed the distribution of landform classes in terms of occupied area and the overlapping of classes with sites known to have low, medium, and high productivity. Based on these observations, we created the hierarchical categories related to productive potential (Table 1). The same process was used for evaluating and creating classes related to the water flow power, which represents the gravity action potential on the water; it was observed that areas of greater flow power tended to have higher productivity. It is important to emphasize that this behavior may revert at very high flow levels or in regions with high erodibility.

At the same time, we used images from the Lansat 8 satellite OLI sensor to create maps referring to the NDVI (normalized difference vegetation index) variation in the three different years, always at the end of April and the beginning of May. This period was chosen because of coffee plants' great physiological activity at this time, providing high values of NDVI. To use the images, we made the radiometric corrections using the Semiautomatic Classification Plugin (SCP) of QGIS, based on the data from the MTL file corresponding to each image, using the DOS1 method [36]. It is important to note that the NDVI index relates the near-infrared and red bands, as shown in Equation (1). The literature shows that the near-infrared band is directly related to the amount of vegetation biomass, and the red band is inversely related to plants' photosynthetic activity and senescence stage [24,37]. In

this way, we consider the NDVI as an indirect measure of the relationship between biomass and photosynthesis, then an adequate index for plant health analysis.

$$NDVI = ((NIR - Red))/((NIR + Red)) \tag{2}$$

where: NIR—Near Infrared band and Red—Red band

From the results obtained from the NDVI and the potential production factor map (T factor), the values of the zonal statistics were extracted for the three years of study (2016, 2018, and 2020), as explained above. Specifically, descriptive measures were obtained for each analyzed stand, such as the mean, the standard deviation, and the minimum and maximum values related to the occurrence of the T factor and NDVI classes. These statistics were obtained using the vector related to the division of the fields on the farm as a layer of zones (1999, 2000, 2002, 2003, 2004, and 2007). This operation was performed for all class maps produced using the average value of classes for each field in each parameter evaluated as a variable value in the equations.

As a contribution of this work to the method of calculating the productive potential of coffee plantations, with the values obtained for T and NDVI, we calculated the final productive potential (Pfinal) and the harvest prediction for each stand (Prod), according to the relationship presented in the Equation (3).

$$Pfinal = NDVI + T \tag{3}$$

We can still put the model in the form of a single equation, which would be (Equation (4)):

$$Pfinal = NDVI + ((Declivity + Temperature)/(Flux + Shape)) \tag{4}$$

*2.3. Analysis of Results*

We analyzed data from 3 consecutive crops (years) of coffee plants installed in the study area. The selected years were the 2016, 2018, and 2020 harvests, and the odd years were interspersed with zero-harvest years. A statistical description of the real data was performed, from which we verified the correlation with the results estimated by the model based on the reading of the landscape variables, according to our adaptation of the Ross model (1994). The equations used in our model were adjusted to obtain an optimal correlation for the year 2018. The model (equations) was replicated unchanged in the other two harvests (2016 and 2020), generating new linear regressions. Specifically, the data obtained for the zonal statistics of all parameters evaluated were analyzed using descriptive statistics. The values obtained from the estimated production potential calculated for the three years in each plot of the study area were compared with the historical data of the farm from the linear regression analysis and, in general, from the RMSE (root mean square error) between the calculated productive potential and the observed yield. The same analyses were performed directly between the NDVI values transformed for production and the observed productivity under linear regression and RMSE and between the T factor (excluding NDVI) and productivity.

## 3. Results and Discussion

The developed model could predict the relationship between the observed productivity and the productive potential calculated in each stand, each year, with a reasonable accuracy percentage. In this sense, we observed a high correlation ($R^2$) for the applied regressions between the calculated productive potential and the observed productivity for the years 2018 and 2020 ($R^2$ 0.91 and 0.93), while for the year 2016, this value was lower (0.85), a still high value. In the first two years, the adjustment was linear, while for 2016, it was a polynomial of the second degree, with a sharp drop in the relationship between the stands with the highest potential and productivity, probably due to the still-existing biennial. These values can be seen in the graphs in Figure 4. The RMSE values show a low standard

deviation among the residuals, with 8.96, 5.3, and 4.47 bags per hectare of RMSE for 2016, 2018, and 2020, respectively.

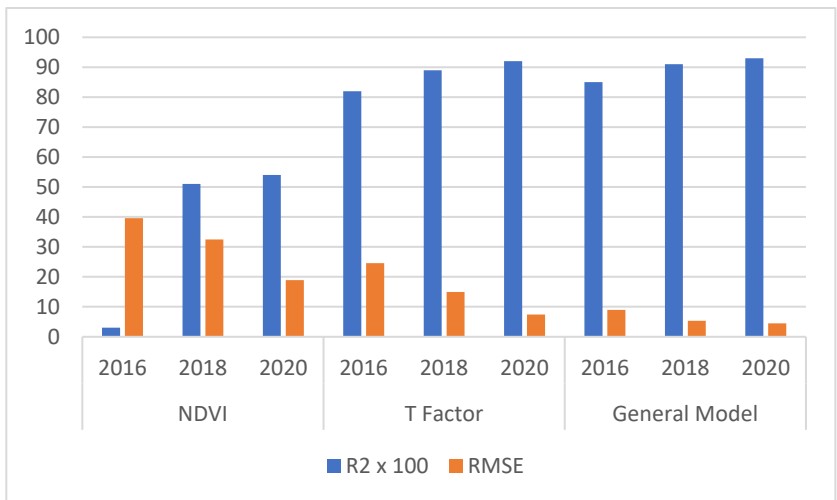

**Figure 4.** $R^2$ values multiplied by 100 (percentage) and RMSE for the 3 comparison models in the 3 different years.

Also, when we compare the model gains regarding the direct comparison of the NDVI or the isolated T factor of the NDVI and the real productivity, we observe a gain in reliability both in the $R^2$ and the RMSE analysis, as seen in Figure 4. While comparing the values of productivity and NDVI, we had an average of 0.36 for $R^2$ and 30.31 bags for RMSE for the 3 years. In contrast, for the comparison between the T factor and productivity, these values were 0.876 and 15.61, and for the general model 0.896 and 6.24, respectively.

It is interesting to note that there is a very high gain in the correlation of NDVI with productivity with the stabilization of the zero-crop system in consortium with Brachiaria, which may be due to the forced directing of the drain to the reproductive caused by the management system, as well as by the high correlation that exists between the NDVI response of C4 plants, such as Brachiaria, to the productive levels of areas and local fertility [38–40]. As for the topography analysis, a high correlation is observed even in the absence of NDVI, corroborating the idea that this factor dramatically influences the productive potential of the areas [14,33,41]. In any case, the addition of NDVI to the model generated an increase in the general precision regarding the analyzed parameters.

Table 2 contais the data relating to the actual yields observed and the descriptive statistics relating to the annual variations of the general average and standard deviation per field and the general average of the farm for the 3 years. Table 3 contains the historical productivity values of the areas obtained from data provided by the property where the study was carried out, as well as the standard deviation and average statistics of the productivity variable per plot and per agricultural year.

**Table 2.** Hierarchical categories for coffee productivity given slope, flow power, and terrain shape factors adapted from Ross's proposition (1994).

| Hierarchical Categories | Declivity (%) | Flow Power (W/m) | Terrain Shape (Topodata Classes) | Temperature (Celsius) |
|---|---|---|---|---|
| Very bad | >25 (1) | 0–2 (1) | 8 and 9 (1) | >20.13 (1) |
| Bad | 16–25 (2) | 2–4 (2) | 6 and 7 (2) | 19.90–20.13 (2) |
| Medium | 12–16 (3) | 4–6 (3) | 4 and 5 (3) | 19.67–19.90 (3) |
| Good | 0–2 and 6–12 (4) | 6–8 (2) | 1 and 3 (2) | 19.44–19.67 (4) |
| Very Good | 3–6 (5) | >8 (1) | 2 (1) | 19.00–19.44 (5) |

Source: The authors.

**Table 3.** Values of observed absolute coffee productivity and mean and standard deviation by year and field.

| Plot | Productivity (Benefited Bags.hectare$^{-1}$) | | | | Standard Deviation (Benefited Bags.hectare$^{-1}$) |
|---|---|---|---|---|---|
| | **2016** | **2018** | **2020** | **Medium** | |
| D | 106 | 101 | 102 | 61.80 | 2.65 |
| C | 102 | 96 | 90 | 57.50 | 5.76 |
| G | 111 | 82 | 88 | 56.17 | 15.23 |
| F | 100 | 80 | 87 | 53.40 | 10.15 |
| E | 101 | 78 | 82 | 52.26 | 12.47 |
| B | 72 | 51 | 76 | 39.73 | 13.36 |
| A | 70 | 54 | 66 | 38.07 | 8.46 |
| Medium | 95 | 77 | 84 | 85.33 | |
| Standard Deviation | 16.49 | 19.04 | 11.37 | | |

We observed less variation in productivity in stands with higher yields, while from stand G (3rd place in production level), there is a greater temporal variability. Also, there is no great variation in the hierarchical positioning of the stands regarding productivity over the years. This reinforces the idea that productivity is linked, among other factors, to some determinants and limiting factors with low temporal variability, such as topography. Also, the average productivity of the farm was higher in 2016, followed by 2020 and 2018. The values observed here are probably due to the determining or limiting dynamic factors that govern productivity, such as the weather, which defines or determines the achievable productivity in one year.

There is a direct correlation between the estimated productive potential (Figure 5) and the hierarchy of topographic and temperature classification of the area (Figures 5–7). Punctually, the most remarkable correspondence occurred between terrain forms (TF) and flow power (FP). Thus, we emphasize the possibility of defining management zones from the topography with specific management [13,33], considering the need to foresee the productive potential of the different areas of rural property. We also observed a tendency for lower surface temperatures in the places of occurrence of the classes of greater ground concavity, which coincide with greater FP.

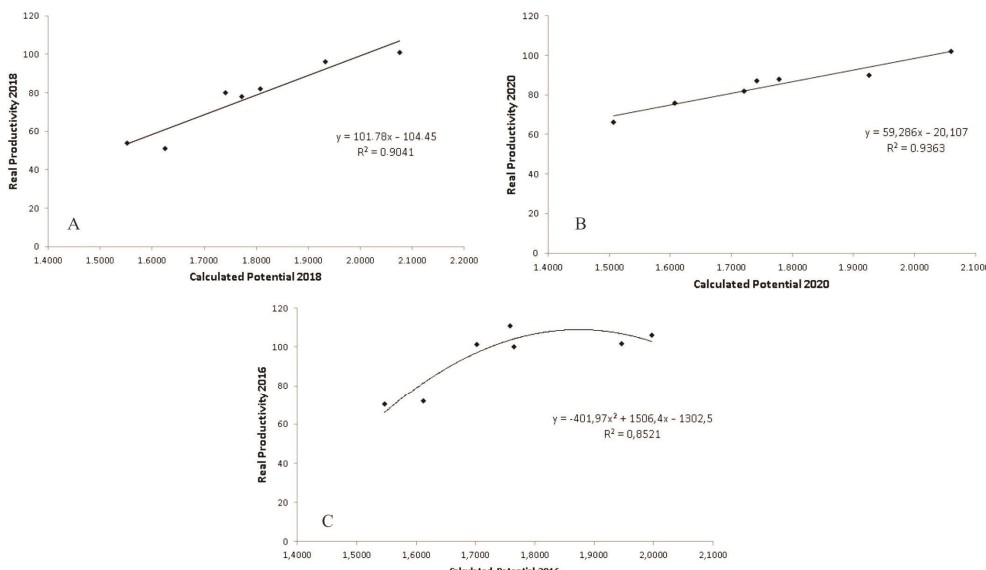

**Figure 5.** Graphs of the regression analysis between actual and potential productivity calculated for the 3 years of study. (**A**)—productive potential in 2018; (**B**)—productive potential in 2020; (**C**)—productive potential in 2016.

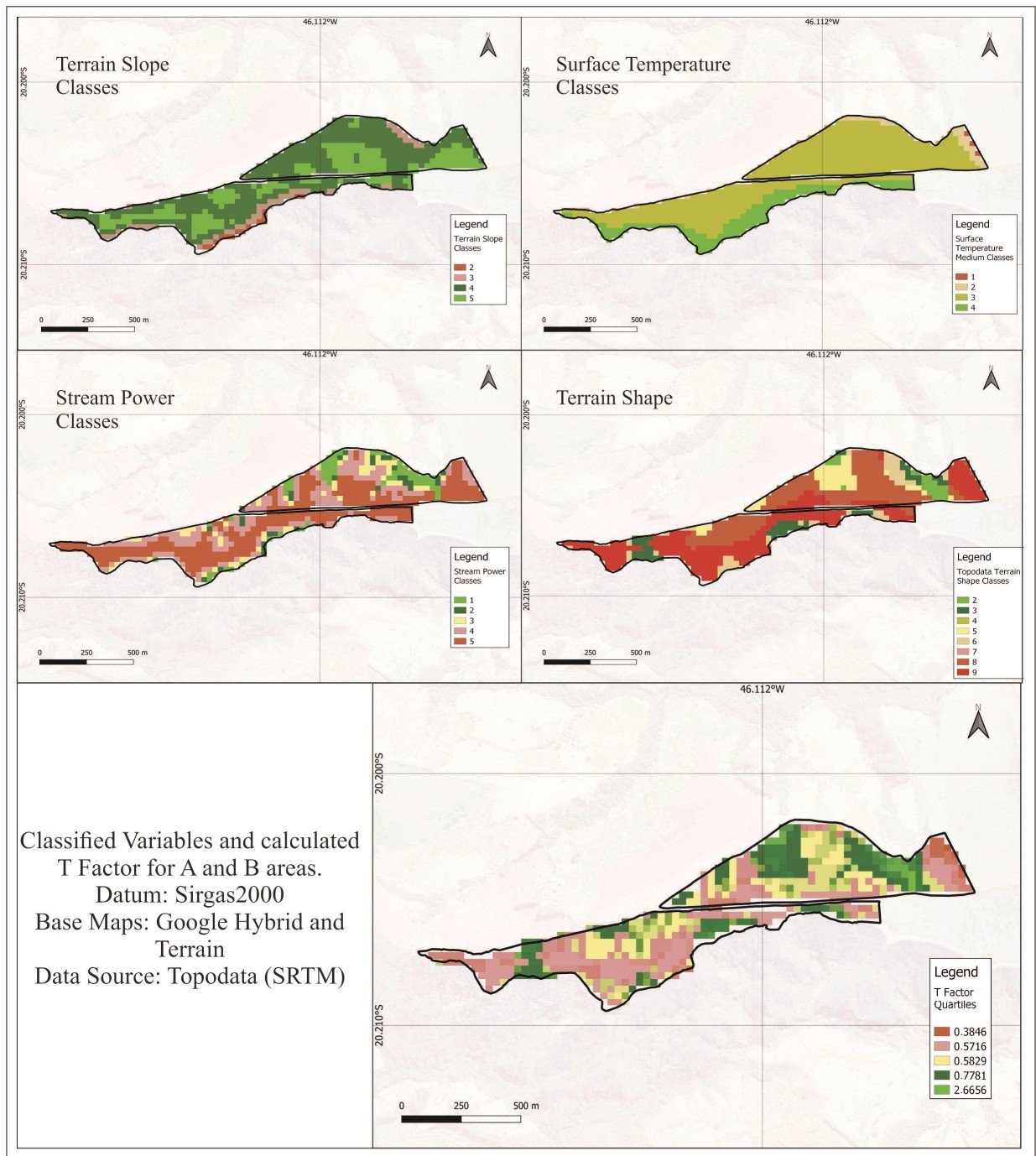

**Figure 6.** Class maps of the variables used and the level of productive potential according to the topography (FactorT) for areas A and B.

When comparing the direct correlations between the variables and productivity using linear regression, the highest correlation was observed with the variable terrain shape, followed by water flow capacity, NDVI, and pure slope (Table 4). It is worth noting that the variation in slope is slight in the study area compared to other coffee-growing regions, where this may be a more influential variable.

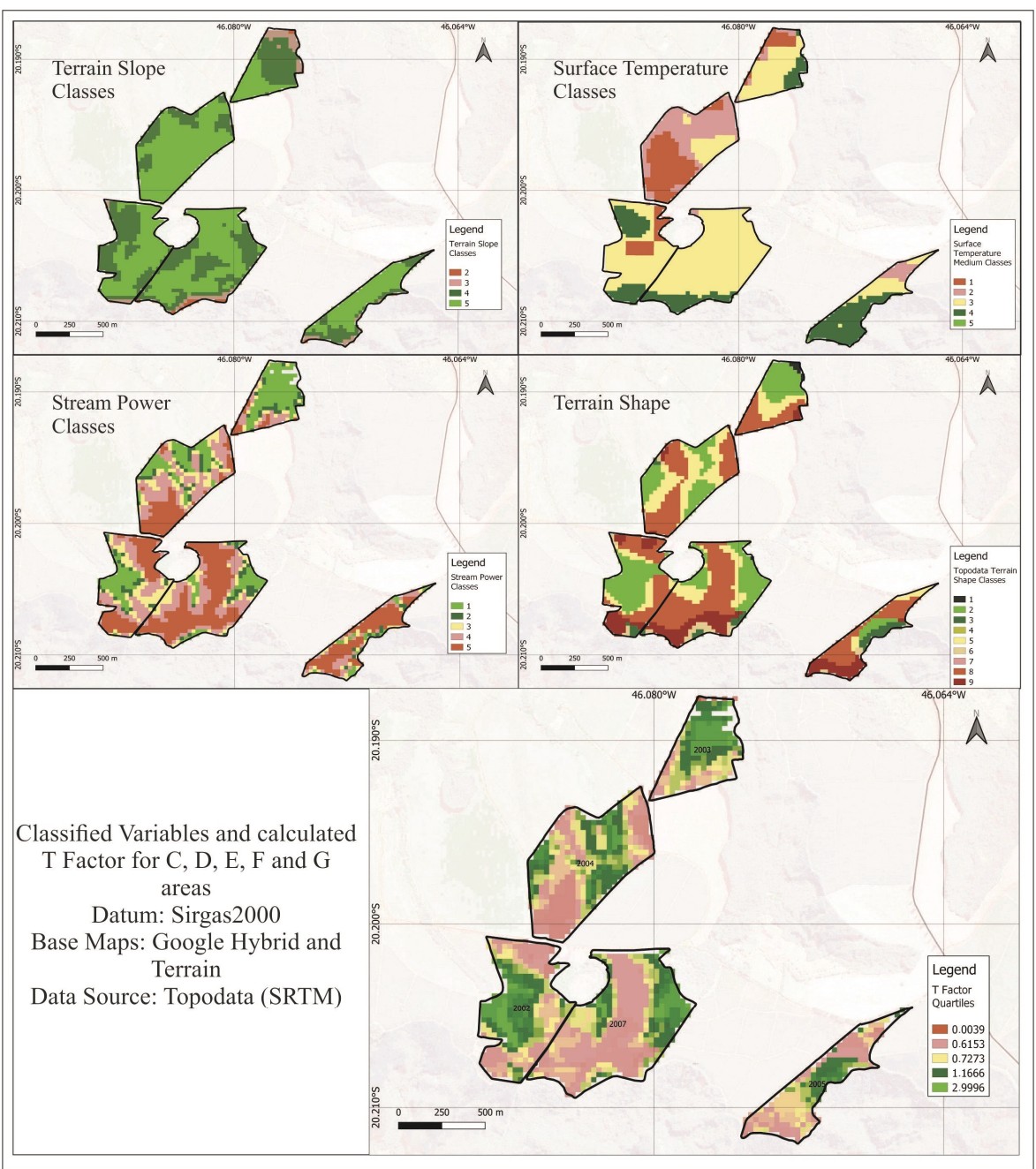

**Figure 7.** Class maps of variables used and the level of productive potential according to topography (FactorT) for areas C, D, E, F, and G.

**Table 4.** Values of observed absolute coffee productivity, mean and standard deviation by year and field.

| Variable | $R^2$ | Equation |
|---|---|---|
| Terrain Shape | 0.7723 | $y = -7.6959x + 98.66$ |
| Water Flow Power | 0.4144 | $y = 8.3527\ln(x) + 35.082$ |
| NDVI | 0.3908 | $y = 303.46x - 206.23$ |
| Slope | 0.1073 | $y = -0.7107x^2 + 7.1845x + 35.273$ |

We note in Table 5 that topography and temperature are factors that initially influence the levels of productive potential. Still, both topography and temperature have a

prominent influence on the vegetative quality of the crop, i.e., the same vegetation index (NDVI) under different locations, with different potentials related to topography or plant temperature, will result in different productivity values. Figures 8 and 9 illustrate the addition behavior between topography and vegetation (NDVI), resulting in a final annual potential for the areas. These results corroborate the idea that topography would be one of the main factors determining the productive potential of a coffee area, influencing the initial potential of sites, as it is directly related to soil parameters [13,14,30,42]. This also implies that terrain variations can be used to predict coffee-crop behavior even within the same stand, highlighting the importance of investments in mapping digital surface models and soil types.

**Table 5.** Data for the medium of different parameter classes observed by pixel, estimated productive potential, and observed productivity (Class Means—Dimensionless).

| Data/Plot | A | B | C | D | E | F | G |
|---|---|---|---|---|---|---|---|
| Terrain Shape | 7.77 | 6.91 | 5.35 | 4.81 | 5.46 | 6.64 | 6.16 |
| Flow Power | 4.29 | 3.7 | 3.41 | 2.06 | 3.36 | 4.08 | 3.95 |
| Declivity | 4 | 4 | 5 | 4 | 5 | 5 | 5 |
| Temperature | 4.45 | 4.49 | 3.28 | 4.68 | 4.39 | 4.23 | 4.66 |
| T Factor | 0.70 | 0.80 | 0.93 | 0.90 | 1.07 | 1.19 | 0.95 |
| NDVI medium 2018 | 0.8514 | 0.8242 | 0.8621 | 0.8788 | 0.8328 | 0.8384 | 0.8522 |
| NDVI medium 2020 | 0.8461 | 0.8120 | 0.8742 | 0.7991 | 0.7640 | 0.8613 | 0.8024 |
| NDVI medium 2016 | 0.8066 | 0.8076 | 0.8539 | 0.8621 | 0.7822 | 0.8388 | 0.8226 |

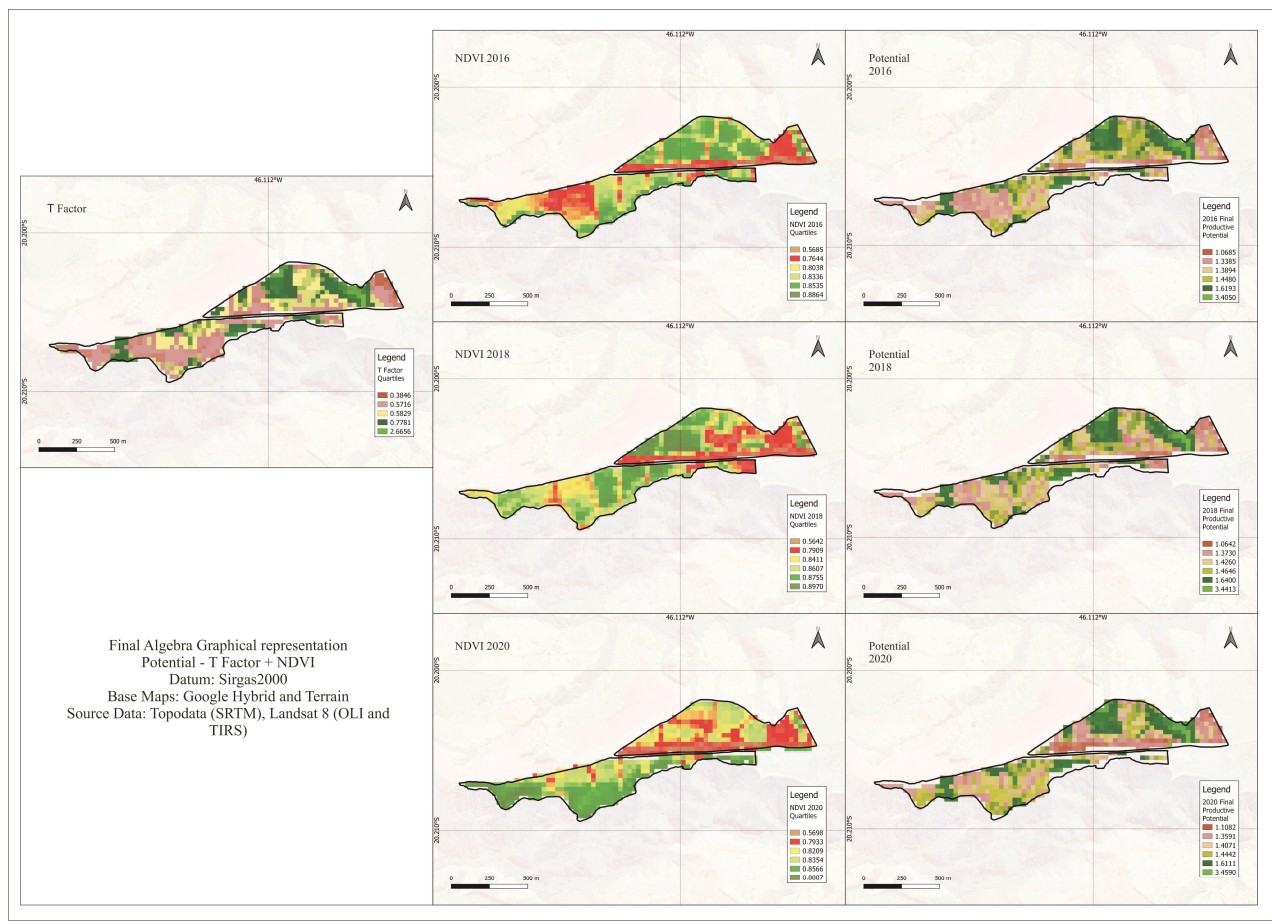

**Figure 8.** Maps of the final productive potential of stands A and B.

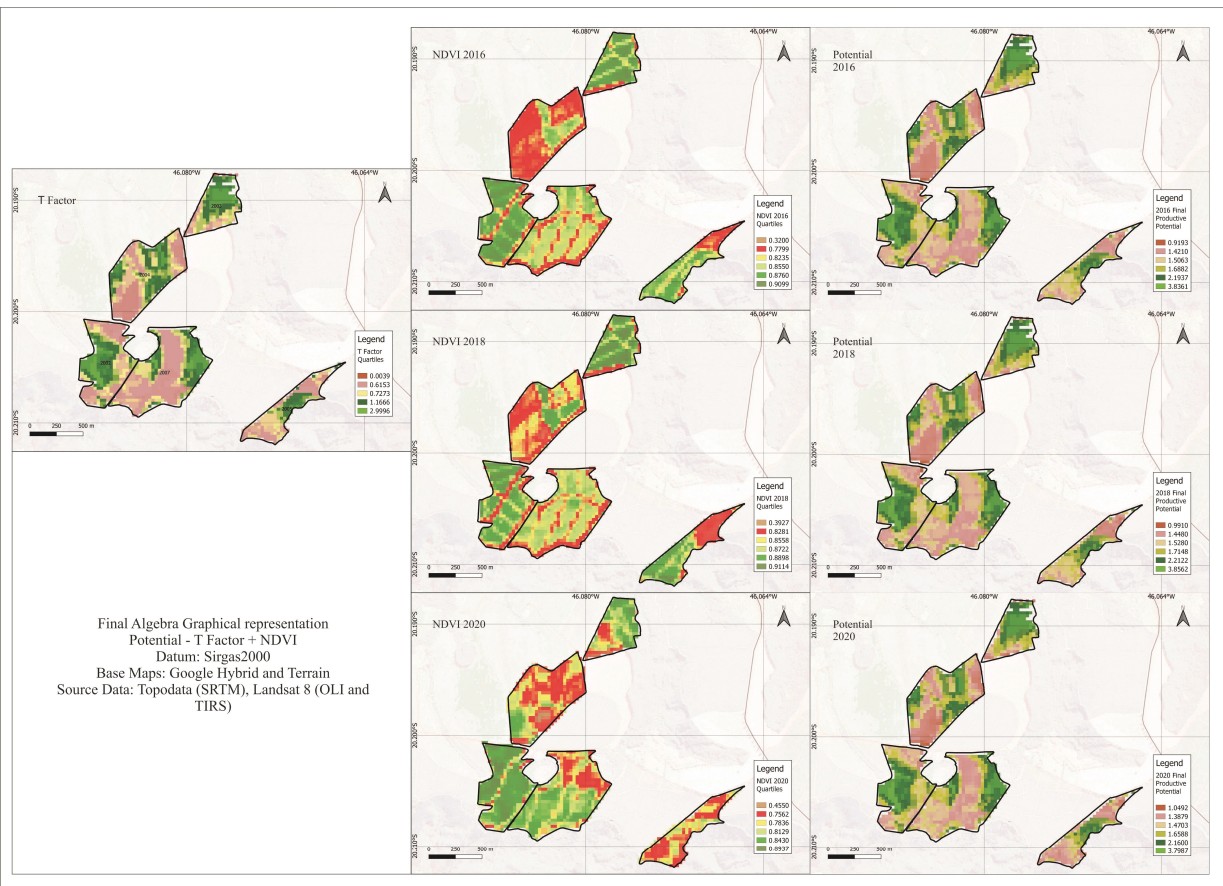

**Figure 9.** Maps of the final productive potential of stands C, D, E, F, and G.

In 2016, the transition from the traditional biennial system to the zero-harvest system happened. We found a decrease in the correlation between the predicted and observed values and a change in the type of correlation between the variables (linear to polynomial), which can be explained by the decrease in plant vigor due to the existing crop in the previous year. Thus, 2016 was the first year the zero-harvest system was introduced, i.e., there was a small harvest in 2015 (low harvest). As a result, sites with slightly better yields in 2015 tended to harvest less in 2016, since there is a considerable loss of plant vigor for the following year [43–45]. Also, adopting the zero-crop system seems to facilitate modeling like the one we created, considering the necessity to observe only one type of harvest for the crop, and there is no adaptation between years of high and low harvests.

Although several works successfully correlate multispectral images and indices to coffee-production parameters [46], the direct correlation with productivity or productive potential is still challenging [47]. It happens mainly due to the need to predict the phenological state of the plant, i.e., the balance between vegetative and reproductive drains that change according to low and high crop years or even between two different high crops [43–45]. In this sense, the high correlation in this model can be attributed to the ease of identifying the drain balance in the zero-harvest system compared to the traditional annual harvest system.

Further, Nogueira et al. (2018) [45] obtained a high level of correlation of indexes with coffee productivity for high harvest years (approximately 89%), while for low harvest years, this value dropped to around 70%, which highlights the better adaptation of models in the high harvest. Our model shows greater precision when we compare the results obtained here with other studies [46]. The gain proposed by our model is the use of the analysis of factors before the quality of the vegetation, in this case, the topography and the age of the



crop, which increases the accuracy of the predictive model regarding the exclusive use of spectral indices derived from the monitoring completed by images.

## 4. Conclusions

It is possible to associate the productive potential of the coffee tree with variables related to topography (slope shapes, water flow power, and slope) and to the plant (age and vigor) to create a prediction model for crop yields. This enables the rational planning of the use of inputs by allowing the allocation of resources according to the area's potential; it is also possible to classify areas of greater potential at the expense of exploring less productive regions, generating productivity allied to environmental conservation.

The model we created proved to be satisfactorily correct in measuring the productive potential of coffee trees in a zero-harvest system. Furthermore, the results indicate that the methodology should be replicated in other regions with coffee plantations, considering the necessary adaptations to local conditions. We emphasize that, under a classical harvest system, that is, annual harvest, the modeling should consider creating different scenarios for intercalated years.

We observed that the topography integrates factors that influence the potential of the areas to the productivity of the coffee plant. In contrast, age and current vegetative vigor impact the dynamics and how the crop expresses the previously defined potential.

In this way, it is possible to use the precepts proposed by Ross (1994) for environmental fragility in an adapted form, according to the reality of each location, for the assembly of predictive productivity models for the coffee crop.

**Supplementary Materials:** The following supporting information can be downloaded at: https://zenodo.org/record/8250683.

**Author Contributions:** Conceptualization, J.E.F.C. and J.V.M.B.; methodology, J.E.F.C., G.D.M. and J.V.M.B.; software, J.E.F.C.; validation, G.D.M., E.F.F.J., S.P.C. and J.V.M.B.; formal analysis, G.D.M., E.F.F.J., S.P.C. and J.V.M.B.; investigation, J.E.F.C.; resources, J.E.F.C.; data curation, J.E.F.C.; writing—original draft preparation, J.E.F.C.; writing—review and editing, G.D.M., E.F.F.J., S.P.C. and J.V.M.B.; supervision, G.D.M. and J.V.M.B.; project administration, J.V.M.B.; funding acquisition, J.V.M.B. All authors have read and agreed to the published version of the manuscript.

**Funding:** This research received no external funding.

**Data Availability Statement:** Statistics, productivity values tables, and Supplementary Materials (figures, graphics, and tables) can be downloaded at: https://zenodo.org/record/8250683 (accessed on 18 September 2023). Topodata files regarding terrain can be downloaded at: http://www.webmapit.com.br/inpe/topodata/ (accessed on 18 September 2023). and Sentinel 2 images can be found at: https://scihub.copernicus.eu/dhus/#/home (accessed on 18 September 2023).

**Conflicts of Interest:** The authors declare no conflict of interest.

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
