# Peer review of "A Multicriteria Model for Estimating Coffea arabica L. Productive Potential Based on the Observation of Landscape Elements"

_agriculture, doi:10.3390/agriculture13112083_

Round 1
Reviewer 1 Report
Comments and Suggestions for Authors
This is an interesting article. However, it needs extensive editing and revision.
The introduction section was poorly structured and needs to be revised. It lacks a proper layout. To me – it looks like a report, not a research article. Since this is a scientific publication, authors have to think about the language. The authors used “we” in many places in the introduction which needs to be changed.
Materials and Methods - Even though the authors add sufficient information, it is not clear and poorly structured. For example, authors can use tables to describe some of the content available in the paragraphs. Due to the lack of proper structure and flaws in the language, it is difficult to understand the content.
The conclusions section was poorly structured and contains unnecessary information. Please align the conclusions with the objectives
Some specific comments are available in the attached pdf.

Comments on the Quality of English LanguageExtensive editing of the English language required
Author Response
We accept the reviewer's suggestions in almost all cases. We have answered the specific questions directly in the PDF. We have improved the manuscript considering the comments of both reviewers. We also paid for checking the English issues. We see the paper has greatly improved after carefully reviewing all the problems. Thank you.

Reviewer 2 Report
Comments and Suggestions for Authors
This study investigated and demonstrated the possibility of monitoring coffee stand productivity using elevation and satellite images. I found the topic interesting. Purely applying existing models in the literature (i.e. Ross model in 1994 L262) is not new. To make this study impactful, I suggest the authors further investigating the importance of NDVI, terrain features, etc. in explaining the productivity potential in a vigorous way.
The authors first built a model (equation) to estimate productivity potential (Equation 4 on L255), demonstrated a strong correlation between the estimated productivity potential and final yield using 7 sites for separate three years, and concluded the model was able to estimate production potential. There are four major concerns in the methodology that the author presented in the manuscript:
-
The authors showed a strong correlation between modeled coffee yield potential and the final observed yield for three years (2016, 2018, and 2020) based on a total of 7 sites. The number of sampling points is too small (7 sites) to conclude that “our model was able to estimate the yearly production potential” (Line 21).
-
The authors seem to use 2018 data to show the strong correlation between productivity potential and yield, and then replicated the correlation analysis for 2016 and 2020 data (L263-264). What’s the reason to choose first fit the correlation for 2018 data? Why the correlation for 2016 is not linear compared to 2018 and 2020 data?
-
The authors mentioned about RMSE (L268) but it’s not clear which was compared to which to calculate the error. If the authors calculated RMSE between productivity potential and final yield, I do not believe the small RMSE means anything because the authors did not use a rigorous procedure to develop yield prediction model in the manuscript.
-
Figure 4 mentioned about 3 models. However, the three models were not explained in section 2.3. Section 2.3 explained one model (i.e. equation 4).
Other comments:
-
In Abstract, I would like to see 1-2 sentences about the implications of your study. I believe the contribution of this study is the reasons for high vs low productivity potential based on the remote sensing products.
-
L81-84: turn these sentences into “the objective of this study …”
-
Add reference for L88-90.
-
What is the equations in the literature L161-162? The authors should present it in the manuscript.
-
Is “T” in equation 1 the same as “Hierarchical categories” in table 1? The authors need to provide clarity in L195-200.
-
How was “Terrain shape” calculated in Table 1? The authors need to explain.
-
What’s the unit for “T” in equation 1 and “Pfinal” in equation 3?
-
Correct “fluxo” and “Forma” in Equation 4.
-
In “Table 2”, did the authors mean “.” when they typed “,”? For example, the medium productivity for plot D is 61,80, is it “6180” or “61.80”? How is medium and standard deviation calculated? If you have more than 7 data points, it is important to include in the result Figure 5.
-
To improve the impact and pulishable aspect of this manuscript, I suggest
-
Add the spatial distribution of estimated “Pfinal” alongside with each variables in the equation (Figure6-8)
-
Statistically demonstrate which variables are more important in determining the
“Pfinal”. -
How was table 3 generated? The caption for table 3 or the explanation in the manuscript (L337) does not tell the readers much.
-
Throughout the manuscript, avoid using informal languages such as “We understand”, “we observed”, etc
-
Throughout the manuscript, avoid using informal languages such as “We understand”, “we observed”, etc
Author Response
- The authors showed a strong correlation between modeled coffee yield potential and the final observed yield for three years (2016, 2018, and 2020) based on a total of 7 sites. The number of sampling points is too small (7 sites) to conclude that "our model was able to estimate the yearly production potential" (Line 21).
There are 7 plots, but the analysis is based on a combination of different data (pixels) per plot and a time series of 3 different years. We agree that we cannot say that this model is valid for other situations, not even within the same area in a different planting system. However, in general lines, we have shown that there are ways of defining the productive potential of areas based on the reading of natural features. Based on your comments, we emphasized in the manuscript that our statements refer only to the study area and within the zero crop system. Additionally, we would like to reinforce that this experimental study considered an area with real data and that the conditions are very similar in the coffee-growing areas of Minas Gerais. In future studies, we intend to validate the model for different areas, but one difficulty is accessing the producers' data; this is sensitive data.
- The authors seem to use 2018 data to show the strong correlation between productivity potential and yield, and then replicated the correlation analysis for 2016 and 2020 data (L263-264). What's the reason to choose first fit the correlation for 2018 data? Why the correlation for 2016 is not linear compared to 2018 and 2020 data?
Thank you for your comment. 2018 was chosen because it was the first year of the farm's fully established zero-crop system. As for the correlation, we explain this behavior (L362-367). 2016 was the first year the zero harvest system was introduced, i.e. there was a small harvest in 2015 (low harvest). As a result, sites with slightly better yields in 2015 are at a disadvantage in 2016, since there is a considerable loss of plant vigor for the following year, when it is normal for these plants to direct their drains to vegetative growth, which results in higher NDVI and lower yields at the same time. This is why we also point out that the model. However, it shows great potential for use in zero-harvest sites or for forming specific management zones that should not have the same response on farms with a traditional biennial harvesting system. We have also improved this paragraph to make it more explicit and added references to sustain this affirmative.
- The authors mentioned about RMSE (L268) but it's not clear which was compared to which to calculate the error. If the authors calculated RMSE between productivity potential and final yield, I do not believe the small RMSE means anything because the authors did not use a rigorous procedure to develop yield prediction model in the manuscript.
Thank you. The analysis was based on observed and calculated yield data (model). We have improved the sentence to make it more transparent. However, we disagree that its use is inappropriate since this metric is commonly used to check for errors within linear, polynomial and quadratic regressions, which were the correlation methods used in this work (e.g. https://doi.org/10.1029/WR026i009p02069)
- Figure 4 mentioned about 3 models. However, the three models were not explained in section 2.3. Section 2.3 explained one model (i.e. equation 4).
Thanks for the note. While setting up the model, we carried out similar analyses using only the NDVI transformed to productivity in the form of a linear regression and the T factor alone, without adding the NDVI to the model. These are the other 2 models we are referring to. NDVI alone showed very little correlation with yield potential, so we no longer present the results in graphic form or images. As for the T factor alone, the model showed a good correlation, slightly lower than the final model, which is why we haven't presented it either. In any case, we've added it to the text so that it's clear which are the other two models.(end of 2.3)
Other comments:
- In Abstract, I would like to see 1-2 sentences about the implications of your study. I believe the contribution of this study is the reasons for high vs low productivity potential based on the remote sensing products.
Thank you. We have changed the introductory part of the abstract.
- L81-84: turn these sentences into "the objective of this study …"
Thanks for the recommendation. We changed in the text.
- Add reference for L88-90.
As requested, references to the sentence have been added. These have been used in other parts of the article.
- What is the equations in the literature L161-162? The authors should present it in the manuscript.
Thanks for pointing that out. We agree on the importance of explaining Ross's method better.
In this case, Ross's proposal was to assign weights to different natural parameters, such as slope, type of vegetation and soil orders, according to empirical observations and to use simple algebra, such as sums and multiplications, to generate a value that functions as an environmental fragility score. This method has been reused with different parameters and different equations by various authors, according to each specific case. In this study, we have adapted the idea to create productive potential scores. This value functions as an admensional class.
We have added a paragraph explaining it as requested.
- Is "T" in equation 1 the same as "Hierarchical categories" in table 1? The authors need to provide clarity in L195-200.
Thank you for your contribution; as proposed, we have added a sentence explaining the values used to calculate T.
- How was "Terrain shape" calculated in Table 1? The authors need to explain.
Thank you, a sentence has been added after figure 3 explaining the processes used within QGIS. The terrain classes in question are offered directly for download on the Topodata-Inpe website, according to Valeriano's classification (2008)
- What's the unit for "T" in equation 1 and "Pfinal" in equation 3?
T is adimensional, according to the previous notes; now we have clarified this point in the text. Pfinal is given in 60 kg bags of coffee per hectare.
- Correct "fluxo" and "Forma" in Equation 4.
Thank you, corrected.
- In "Table 2", did the authors mean "." when they typed ","? For example, the medium productivity for plot D is 61,80, is it "6180" or "61.80"? How is medium and standard deviation calculated? If you have more than 7 data points, it is important to include in the result Figure 5.
Thanks for the correction. The original text was in Portuguese and we failed to replace the comma with a period (it is used here). We've also corrected table 3, which had the same problem. The average and standard deviation have been calculated on the productivity values, given in bags per hectare. As for figure 5, the result was subdivided between figures 5 and 6 due to the visualization of the images. Visualization would be greatly compromised as the plots are a little far apart.
- To improve the impact and pulishable aspect of this manuscript, I suggest
- Add the spatial distribution of estimated "Pfinal" alongside with each variables in the equation (Figure6-8)
Thank you. Note that Pfinal is a combination of the T factor and the NDVI values, it appears in the last quadrant of figures 8 and 9, as a combination of the NDVI values and the T factor.
- How was table 3 generated? The caption for table 3 or the explanation in the manuscript (L337) does not tell the readers much.
Thank you; it was poorly explained. The data refers to the average obtained from pixel-by-pixel classes within each plot observed. We've changed the title to:
Data for the medium of different parameter classes observed by pixel, estimated productive potential, and observed productivity (Class Means – Dimensionless).

Round 2
Reviewer 1 Report
Comments and Suggestions for Authors
Thank you for the reminder. I went through the revised version. The authors revised the paper according to the comments.
Reviewer 2 Report
Comments and Suggestions for Authors
the authors have addressed my comments. The manuscript is ready for publication